# TASK-AGNOSTIC GRAPH EXPLANATIONS

## ABSTRACT

Graph Neural Networks (GNNs) have emerged as powerful tools to encode graph structured data. Due to their broad applications, there is an increasing need to develop tools to explain how GNNs make decisions given graph structured data. Existing learning-based GNN explanation approaches are task-specific in training and hence suffer from crucial drawbacks. Specifically, they are incapable of producing explanations for a multitask prediction model with a single explainer. They are also unable to provide explanations in cases where the GNN is trained in a self-supervised manner, and the resulting representations are used in future downstream tasks. To address these limitations, we propose a Task-Agnostic GNN Explainer (TAGE) trained under self-supervision with no knowledge of downstream tasks. TAGE enables the explanation of GNN embedding models without downstream tasks and allows efficient explanation of multitask models. Our extensive experiments show that TAGE can significantly speed up the explanation efficiency by using the same model to explain predictions for multiple downstream tasks while achieving explanation quality as good as or even better than current state-of-the-art GNN explanation approaches.

## 1 INTRODUCTION

Graph neural networks (GNNs) (Kipf & Welling, 2017; Veličković et al., 2018; Xu et al., 2019) have achieved remarkable success in learning from real-world graph-structured data due to their unique ability to capture both feature-wise and topological information. Extending their success, GNNs are widely applied in various research fields and industrial applications including quantum chemistry (Gilmer et al., 2017), drug discovery (Wu et al., 2018; Wang et al., 2020), social networks (Fan et al., 2019), and recommender systems (Ying et al., 2018). While multiple approaches have been proposed and studied to improve GNN performance, GNN explainability is an emerging area and has a smaller body of research behind it. Recently, explainability has gained more attention due to an increasing desire for GNNs with more security, fairness, and reliability. Being able to provide a good explanation to a GNN prediction increases model reliability and reduces the risk of incorrect predictions, which is crucial in fields such as molecular biology, chemistry, fraud detection, etc.

Existing methods adapting the explanation methods for convolutional neural networks (CNNs) or specifically designed for GNNs have shown promising explanations on multiple types of graph data. A recent survey (Yuan et al., 2020) categorizes existing explanation methods into gradient-based, perturbation, decomposition, and surrogate methods. In particular, perturbation methods involve learning or optimization (Ying et al., 2019; Luo et al., 2020; Yuan et al., 2021; Lin et al., 2021) and, while bearing higher computational costs, generally achieve state-of-the-art performance in terms of explanation quality. These methods train *post-hoc* explanation models on top of the prediction model to be explained. Earlier approaches like GNNExplainer (Ying et al., 2019) require training or optimizing an individual explainer for each data instance, i.e., a graph or a node to be explained. In contrast, PGExplainer (Luo et al., 2020) performs inductive learning, i.e., it only requires a one-time training, and the explainer can be generalized to explain all data instances without individual optimization. Compared to other optimization-based explanation methods, PGExplainer significantly improves the efficiency in terms of time cost without performance loss by learning.

However, even state-of-the-art explanation methods like PGExplainer are still task-specific at training and hence suffer from two crucial drawbacks. First, current methods are inefficient in explaining multitask prediction for graph-structured data. For example, one may need to predict multiple chemical properties in drug discovery for a molecular graph. In particular, ToxCast from MoleculeNet has

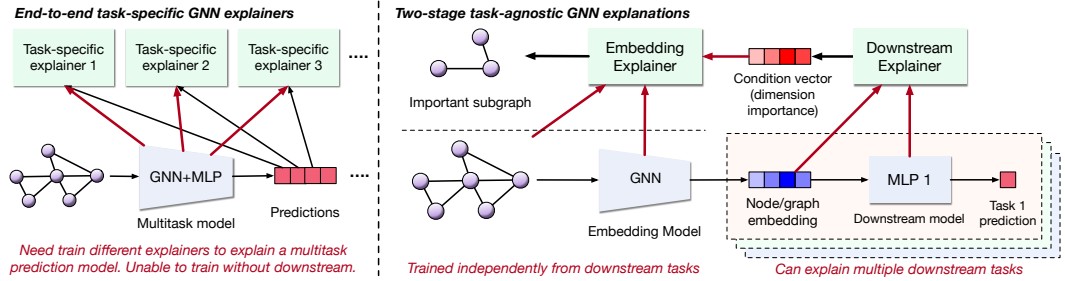

Figure 1: A comparison between typical end-to-end task-specific GNN explainers and the proposed task-agnostic explanation pipeline. To explain a multitask model, typical explanation pipelines need to optimize multiple explainers, whereas the two-stage explanation pipeline only learns one embedding explainer that can cooperate with multiple lightweight downstream explainers.

167 prediction tasks. In these cases, it is common to apply a single GNN model with multiple output dimensions to make predictions for all tasks. However, one is unable to employ a single explainer to explain the above model, since current explainers are trained specifically to explain one prediction task. As a result, in the case of ToxCast, one must train 167 explainers to explain the GNN model. Second, in industry settings, it is common to train GNN models in a two-stage fashion due to scaling, latency, and label sparsity issues. The first stage trains a GNN-based embedding model with a massive amount of unlabeled data in an unsupervised manner to learn embeddings for nodes or graphs. The second stage trains lightweight models such as multilayer perceptrons (MLPs) using the frozen embeddings as input to predict the downstream tasks. In the first stage, the downstream tasks are usually unknown or undefined, and existing task-specific explainers cannot be applied. Also, there can be tens to hundreds of downstream tasks trained on these GNN embeddings, and training a separate explainer for each task is undesirable and downright impossible.

To address the above limitations, we present a new task-agnostic explanation pipeline, shown in Figure 1, where we decompose a prediction model into a GNN embedding model and a downstream model, designing separate explainers for each component. We design the downstream explainers to cooperate with the embedding explainer. The embedding explainer is trained using a self-supervised training framework, which we dub Task-Agnostic GNN Explainer (TAGE), with no knowledge of downstream tasks, models, or labels. In contrast to existing explainers, the learning objective for TAGE is computed at the graph or node embeddings without involving task-related predictions. In addition to eliminating the need for downstream tasks in TAGE, we argue that the self-supervision performed on the embeddings can bring additional performance boost in terms of the explanation quality compared to existing task-specific baselines such as GNNExplainer and PGExplainer.

We summarize our contributions as follows: 1) We introduce the task-agnostic explanation problem and propose a two-stage explanation pipeline involving an embedding explainer and a downstream explainer. This enables the explanation of multiple downstream tasks with a single embedding explainer. 2) We propose a self-supervised training framework TAGE, which is based on conditioned contrastive learning to train the embedding explainer. TAGE requires no knowledge of downstream tasks. 3) We perform experiments on real-world datasets and observe that TAGE outperforms existing learning-based explanation baselines in terms of explanation quality, universal explanation ability, and the time required for training and inference.

## 2 TASK-AGNOSTIC EXPLANATIONS

### 2.1 NOTATIONS AND LEARNING-BASED GNN EXPLANATION

Our study considers the attributed graph $G$ with node set $V$ and edge set $E$. We formulate the attributed graph as a tuple of matrices $(\boldsymbol{A}, \boldsymbol{X})$, where $\boldsymbol{A} \in \{0,1\}^{|V| \times |V|}$ denotes the adjacency matrix and $\boldsymbol{X} \in \mathbb{R}^{|V| \times d_f}$ denotes the feature matrix with feature dimension of $d_f$. We assume that the prediction model $F$ that is to be explained operates on graph-structured data through two components: a GNN-based embedding model and lighter downstream models. Denoting the input space by $\mathcal{G}$, a node-level embedding model $\mathcal{E}_n : \mathcal{G} \to \mathbb{R}^{|V| \times d}$ takes a graph as input and computes

Table 1: Comparisons on properties of common GNN explainers. Inductivity and task-agnosticism are inapplicable for gradient/rule-based methods as they do not require learning. In the last column, we show the number of required explainers for a dataset with $N$ samples and $M$ tasks.

| | Learning | Inductive | Task-agnostic | # explainers required |
|---|---|---|---|---|
| Gradient- & Rule-based | No | - | - | 1 |
| GNNExplainer (Ying et al., 2019) | Yes | No | No | $M * N$ |
| SubgraphX (Yuan et al., 2021) | Yes | No | No | $M * N$ |
| PGExplainer (Luo et al., 2020) | Yes | Yes | No | $M$ |
| Task-agnostic explainers | Yes | Yes | Yes | 1 |

embeddings of dimension $d$ for all nodes in the graph, whereas a graph-level embedding model $\mathcal{E}_g : \mathcal{G} \to \mathbb{R}^{1 \times d}$ computes an embedding for the input graph. Subsequently, the downstream model $\mathcal{D} : \mathbb{R}^d \to \mathbb{R}$ computes predictions for the downstream task based on the embeddings.

Typical GNN explainers consider a task-specific GNN-based model as a complete unit, *i.e.*, $F := \mathcal{D} \circ \mathcal{E}$. Given a graph $G$ and the GNN-based model $F$ to be explained , our goal is to identify the subgraph $G_{sub}$ that contributes the most to the final prediction made by $F$. In other words, we claim that a given prediction is made because $F$ captures crucial information provided by some subgraph $G_{sub}$. The learning-based (or optimization-based) GNN explanation employs a parametric explainer $\mathcal{T}_\theta$ associated with the GNN model $F$ to compute the subgraph $G_{sub}$ of the given graph data. Concretely, the explainer $\mathcal{T}_\theta$ computes the importance score for each node or edge, denoted as $w_i$ or $w_{ij}$, or masks for node attributes denoted as $m$. It then selects the subgraph $G_{sub}$ induced by important nodes and edges, *i.e.*, whose scores exceed a threshold $t$, and by masking the unimportant attributes. In our study, we follow Luo et al. (2020), focusing on the importance of edges to provide explanations to GNNs. Formally, we have $G_{sub} := (V, E_{sub}) = \mathcal{T}_\theta(G)$, where $E_{sub} = \{(v_i, v_j) : (v_i, v_j) \in E, w_{ij} \geq t\}$.

## 2.2 Task-Agnostic Explanations

As introduced in Section 1, all existing learning-based or optimization-based explanation approaches are task-specific and hence suffer from infeasiblity or inefficiency in many real-application scenarios. In particular, they are of limited use when downstream tasks are unknown or undefined and fail to employ a single explainer to explain a multitask prediction model.

To enable the explanation of GNNs in two-stage training and multitask scenarios, we introduce a new explanation paradigm called the task-agnostic explanation. The task-agnostic explanation considers a whole prediction model as an embedding model followed by any number of downstream models. It focuses on explaining the embedding model regardless of the number or the existence of downstream models. In particular, the task-agnostic explanation trains only one explainer $\mathcal{T}_\theta^{(tag)}$ to explain the embedding model $\mathcal{E}$, which should satisfy the following features. First, given an input graph $G$, the explainer $\mathcal{T}_\theta^{(tag)}$ should be able to provide different explanations according to specific downstream tasks being studied. Table 1 compares the properties of common GNN explanation methods and the desired task-agnostic explainers in multitask scenarios. Second, the explainer $\mathcal{T}_\theta^{(tag)}$ can be trained when only the embedding model is available, *e.g.*, at the first stage of a two-stage training paradigm, regardless of the presence of downstream tasks. When downstream tasks and models are unknown, $\mathcal{T}_\theta^{(tag)}$ can still identify which components of the input graph are important for certain embedding dimensions of interest.

## 3 The TAGE Framework

Our explanation framework TAGE follows the typical scheme of GNN explanation introduced in the previous section. It provides explanations by identifying important edges in a given graph and removing the edges that lead to significant changes in the final prediction. Specifically, the goal of the TAGE is to predict the importance score for each edge in a given graph. Different from existing methods, the proposed TAGE breaks down typical end-to-end GNN explainers into two components. We now provide general descriptions and detailed formulations to the proposed framework.

### 3.1 TASK-AGNOSTIC EXPLANATION PIPELINE

Following the principle of the desired task-agnostic explanations, we introduce the task-agnostic explanation pipeline, where a typical explanation procedure is performed in two steps. In particular, we decompose the typical end-to-end learning-based GNN explainer into two parts: the embedding explainer $\mathcal{T}_{\mathcal{E}}$ and the downstream explainer $\mathcal{T}_{down}$, corresponding to the two components in the two-stage training and prediction procedure. We compare the typical explanation pipeline and the two-stage explanation pipeline in Figure 1. The embedding explainer and downstream explainers can be trained or constructed independently from each other. In addition, the embedding explainer can cooperate with any downstream explainers to perform end-to-end explanations on input graphs.

The downstream explainer aims to explain task-specific downstream models. As downstream models are usually lightweight MLPs, we simply adopt gradient-based explainers for downstream explainers without training. The downstream explainer takes a downstream model and the graph or node embedding vector as inputs and computes the importance score of each dimension on the embedding vector. The importance scores then serve as a condition vector input to the embedding explainer. Given the condition vector, the embedding explainer explains the GNN-based embedding model by identifying an important subgraph from the input graph data. In other words, given different condition vectors associated with different downstream tasks or models, the embedding explainer can provide corresponding explanations for the same embedding model. Formally, we denote the downstream explainer for models from $\mathscr{D}$ by $\mathcal{T}_{down} : \mathscr{D} \times \mathbb{R}^d \to \mathbb{R}^d$, which maps input models and embeddings into importance scores $\boldsymbol{m}$ for all embedding dimensions. We denote the embedding explainer associated with the embedding model $\mathcal{E}$ by $\mathcal{T}_{\mathcal{E}} : \mathbb{R}^d \times \mathcal{G} \to \mathcal{G}$, which maps a given graph into a subgraph of higher importance, conditioned on the embedding dimension importance $\boldsymbol{m} \in \mathbb{R}^d$.

The training procedures of the embedding explainer are independent of downstream tasks or downstream explainers. In particular, the downstream explainer is obtained from the downstream model only, and the training of the embedding explainer only requires the embedding model and the input graphs. As downstream models are usually constructed as stacked fully connected (FC) layers and the explanation of FC layers has been well studied, our study mainly focuses on the non-trivial training procedure and design of the embedding explainer.

### 3.2 TRAINING EMBEDDING EXPLAINER UNDER SELF-SUPERVISION

A straightforward idea of explaining an embedding model with no knowledge of downstream tasks is to employ existing explainers and perform explanation on the pretext task, such as graph reconstruction (Kipf & Welling, 2016) or context prediction (Hu et al., 2020), used during the pre-training of GNNs. However, such explanations cannot generalize to future downstream tasks as there are limited dependencies between the pretext task and downstream tasks. Therefore, training an embedding explainer without downstream models or labels is challenging, and it is desirable to develop a generalizable training approach for the embedding explainer. To this end, we propose a self-supervised learning framework for the embedding explainer.

The learning objective of the proposed framework seeks to maximize the mutual information (MI) between two embeddings with certain dimensions masked, i.e., one of the given graphs and one of the corresponding subgraph of high importance induced by the explainer. We introduce a masking vector $\boldsymbol{p} \in \mathbb{R}^d$ to indicate specific dimensions of embeddings on which to maximize MI. During explanation, we obtain the masking vector from the importance vector computed by any downstream explainer $\mathcal{T}_{down}$. As no downstream importance vector is available at training, we sample the masking vector $\boldsymbol{p}$ from a multivariate Laplace distribution due to the sparse gradient assumption, *i.e.*, only a few dimensions are of high importance. Formally, the MI-based learning objective is

$$\max_{\theta} \mathrm{E}_{\boldsymbol{p}}[\mathbf{MI}(\boldsymbol{p} \otimes \mathcal{E}(G), \boldsymbol{p} \otimes \mathcal{E}(\mathcal{T}_{\theta}(\boldsymbol{p}, G)))], \qquad (1)$$

where $\mathbf{MI}(\cdot, \cdot)$ computes the mutual information between two random vectors, $\boldsymbol{p}$ denotes the random masking vector sampled from a certain distribution, $\mathcal{T}_{\theta}(\boldsymbol{p}, G)$ computes the subgraph of high importance, and $\otimes$ denotes the element-wise multiplication, which applies masking to the embeddings $\mathcal{E}(\cdot)$. Figure 2 outlines the training framework and objective. Intuitively, given an input graph and the desired embedding dimensions to be explained, the explainer $\mathcal{T}_{\theta}$ predicts the subgraph whose

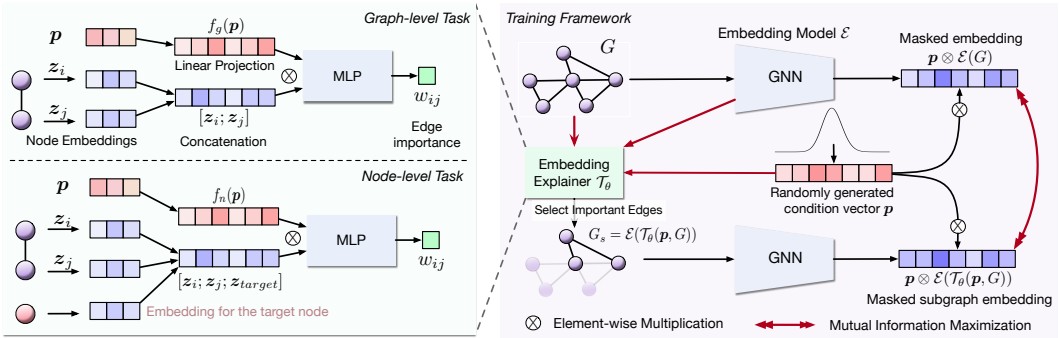

Figure 2: Overviews of the self-supervised training framework for the embedding model (right) and the architecture of the parametric explainers (left). During training, we generate random condition vectors $\boldsymbol{p}$ as an input to the embedding explainer and mask the embeddings. The learning objective seeks to maximize the mutual information between two embeddings on certain dimensions.

embedding shares the maximum mutual information with the original embedding on the desired dimensions.

Practically, the mutual information is intractable and is hence hard to directly compute. A common approach to achieve efficient computation and optimization is to adopt the upper bound estimations of mutual information, namely, the Jenson-Shannon Estimator (JSE) (Nowozin et al., 2016) and the InfoNCE (Gutmann & Hyvärinen, 2010). These upper bound estimations are also referred to as contrastive loss and are widely applied in self-supervised representation learning (Hjelm et al., 2019; Sun et al., 2019; Veličković et al., 2019) for both images and graphs. Adopting these estimators, the objectives are efficiently computed as

$$\min_{\theta} \frac{1}{N} \sum_{i=1}^{N} \log \left[ \sigma \big( (\boldsymbol{p} \otimes \boldsymbol{z}_i)^T (\boldsymbol{p} \otimes \boldsymbol{z}_{i,\theta}) \big) \right] + \frac{1}{N^2 - N} \sum_{i \neq j} \log \left[ 1 - \sigma \big( (\boldsymbol{p} \otimes \boldsymbol{z}_i)^T (\boldsymbol{p} \otimes \boldsymbol{z}_{j,\theta}) \big) \right],$$

$$(2)$$

$$\min_{\theta} -\frac{1}{N} \sum_{i=1}^{N} \left[ \log \frac{\exp\{(\boldsymbol{p} \otimes \boldsymbol{z}_i)^T (\boldsymbol{p} \otimes \boldsymbol{z}_{i,\theta})\}}{\sum_{j \neq i} \exp\{(\boldsymbol{p} \otimes \boldsymbol{z}_i)^T (\boldsymbol{p} \otimes \boldsymbol{z}_{j,\theta})\}} \right], \qquad (3)$$

for JSE and InfoNCE, respectively, where $N$ denotes the number of samples in a mini-batch, $\sigma$ denotes Sigmoid function, $\boldsymbol{z}_i$ and $\boldsymbol{z}_{i,\theta}$ are embeddings of the original graph $G_i$ and its subgraph $\mathcal{T}_{\theta}(G_i)$, or target nodes of the two graphs. Our objective involves condition vectors as masks on the embeddings, which differs from typical contrastive loss used in self-supervised representation learning. We hence call the proposed objective the conditioned contrastive loss.

To restrict the size of subgraphs given by the explainer, we additionally add a size regularization term $R$, computed as the averaged importance score, to the above objectives. In the case where edge importance scores $w_{ij} \in [0, 1]$ are computed, the regularization term is computed as

$$R(G) = \sum_{(v_i, v_j) \in E} \lambda_s |w_{ij}| - \lambda_e \left[ w_{ij} \log w_{ij} - (1 - w_{ij}) \log(1 - w_{ij}) \right], \qquad (4)$$

where $\lambda_s$ and $\lambda_e$ are hyper-parameters controlling the size and the entropy of edge importance scores, respectively.

### 3.3 EXPLAINER ARCHITECTURES

**Embedding explainers**. Inspired by explainer architectures used by PGExplainer, we adopt the multilayer perceptron (MLP) to predict the importance score $w_{ij}$ for each edge $(u_i, u_j) \in E$, on top of learned embeddings $\boldsymbol{z}_i$ and $\boldsymbol{z}_j$ of the two nodes connected by the edge. Edges with scores higher than a threshold are considered as important edges that remain in the selected subgraph. In order for the embedding explainer to cooperate with different downstream explainers and provide diverse explanations for different tasks, it additionally requires a condition vector as input indicating

the specific downstream task to be explained. We handle the condition vector in a similar manner to Conditional GAN (Mirza & Osindero, 2014). Formally, the graph-level embedding explainer takes the embeddings, $z_i$ and $z_j$, and the condition vector $p$ as inputs and computes the importance score by

$$w_{ij} = \text{MLP}_g\big([z_i; z_j] \otimes \sigma(f_g(p))\big), \tag{5}$$

where $[\cdot; \cdot]$ denotes the concatenation along the feature dimension, $\otimes$ denotes the element-wise multiplication, $\sigma$ denotes the activation function, and $f_g : \mathbb{R}^d \to \mathbb{R}^{2d}$ is a linear projection. The node-level embedding explainer takes an additional node embedding as its input, as the explainers are expected to predict different scores for the same edge when explaining different target nodes. The formulation of computing the importance score is as follows,

$$w_{ij} = \text{MLP}_n\big([z_i; z_j; z_{target}] \otimes \sigma(f_n(p))\big), \tag{6}$$

where $f_g : \mathbb{R}^d \to \mathbb{R}^{3d}$ is a linear projection, and $z_{target}$ denotes the embedding of the target node whose prediction is to be explained.

**Downstream explainers**. We adopt the gradient-based explainer to explain the downstream models. Formally, given an input embedding $z$ and its prediction probabilities $\mathcal{D}(z) \in [0,1]^C$ among all $C$ classes, we compute the gradient of the maximal probability w.r.t. the input embedding:

$$g = \frac{\partial \max_{c \leq C} \mathcal{D}(z)[c]}{\partial z} \in \mathbb{R}^{1 \times d},$$

where $\mathcal{D}(z)[c]$ denotes the probability for class $c$. To convert the gradient into the condition vector, we further perform normalization and only take positive values reflecting only positive influence to the predicted class probability, *i.e.*, $p = \text{ReLU}(\text{norm}(g^T))$.

# 4 EXPERIMENTAL STUDIES

We conduct two groups of quantitative studies evaluating the explanation quality and the universal explanation ability, *i.e.*, training a single explainer to explain all downstream tasks, of TAGE. We then compare the efficiency of multiple learning-based GNN explainers in terms of training and explanation time cost. We further provide visualizations to demonstrate the explanation quality as well as the ability to explain GNN models without downstream tasks.

## 4.1 DATASETS

To demonstrate the effectiveness of the proposed TAGE on both node-level and graph-level tasks, we evaluate TAGE on three groups of real-world datasets that contain potentially multiple tasks. The datasets are described as follows and their statistics are summarized in Appendix A.

**MoleculeNet**. The MoleculeNet (Wu et al., 2018) library provides a collection of molecular graph datasets for the prediction of different molecule properties. In a molecular graph, each atom in the molecule is considered as a node, and each bond is considered as an edge. The prediction of molecule properties is a graph-level task. We include three graph classification tasks from MoleculeNet to evaluate the explanation of graph-level tasks: HIV, SIDER, and BACE.

**Protein-Protein Interaction**. The Protein-Protein Interaction (PPI) (Zitnik & Leskovec, 2017) dataset documents the physical interactions between proteins in 24 different human tissues. In PPI graphs, each protein is considered as a node with its motif and immunological features, and there is an edge between two proteins if they interact with each other. Each node in the graphs has 121 binary labels associated with different protein functions. As different protein functions are not exclusive to each other, the prediction of each protein function is considered an individual task. We utilize the first five out of 121 tasks to evaluate the explanation of node-level tasks.

**E-commerce Product Network**. The E-commerce Product Network (EPN)[1] is constructed with subsampled, anonymized logs from an e-commerce store, where entities including buyers, products, merchants, and reviews are considered as nodes, and interactions between entities are considered as edges. We subsample the data for the sake of experimental evaluations and the dataset characteristics

---

[1]Proprietary dataset

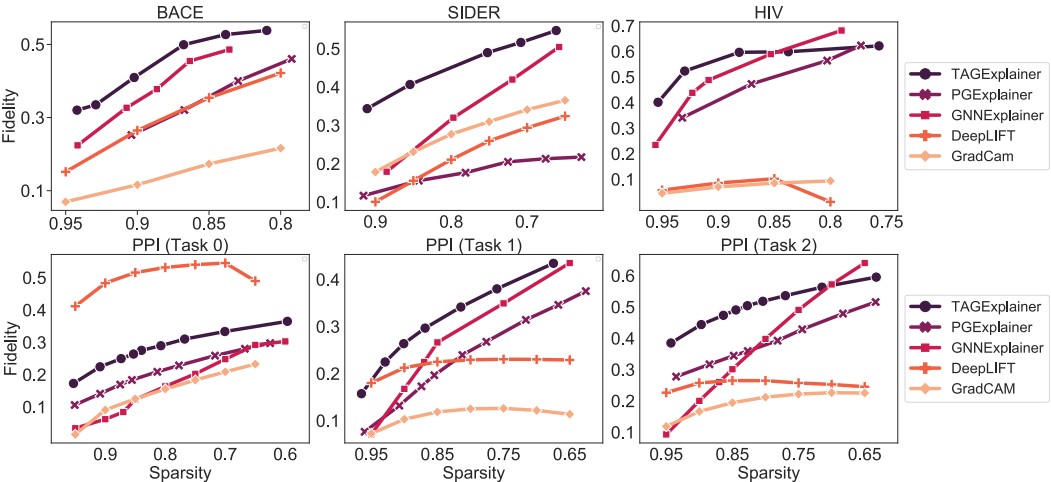

Figure 3: Quantitative performance comparisons with baseline methods on six tasks from MoleculeNet (top row) and PPI (bottom row). The curves are obtained by varying the threshold for selecting important edges.

do not mirror actual production traffic. We study the explanation of the classification of fraudulent entities (nodes), where the prediction for different types of entities are considered as individual tasks. We evaluate our framework specifically on classifications of the buyer, merchant, and review nodes.

## 4.2 EXPERIMENT SETTINGS AND EVALUATION METRICS

For each real-world dataset, we evaluate explainers on multiple downstream tasks that share a single embedding model. For consistency with industrial use cases, we perform the two-stage training paradigm to obtain GNN models to be explained. In particular, we first use unlabeled graphs to train the GNN-based embedding model in an unsupervised fashion. We then freeze the embedding model and use the learned embeddings to train individual downstream models structured as 2-layer MLPs. Specifically, for graph-level classification tasks in MoleculeNet, we employ the GNN pretraining strategy context prediction (Hu et al., 2020) to train a 5-layer GIN (Xu et al., 2019) as the embedding model on ZINC-2M (Sterling & Irwin, 2015) containing 2 million unlabeled molecules. For the node-level classification on PPI, we employ the self-supervised training method GRACE (Zhu et al., 2020) to train a 2-layer GCN (Kipf & Welling, 2017) on all 21 graphs from PPI without using labels. For the larger-scale node-level classification on EPN, we use graph autoencoder (GAE) (Kipf & Welling, 2016) to train the embedding model on sampled subgraphs of EPN. More implementation details are provided in Appendix B.

As the involved real-world datasets do not have ground truth for explanations, we follow previous studies (Pope et al., 2019; Yuan et al., 2020; 2021) to adopt a fidelity score and a sparsity score to quantitatively evaluate the explanations. Intuitively, the fidelity score measures the level of change in the probability of the predicted class when removing important nodes or edges, whereas the sparsity score measures the relative amount of important nodes or nodes associated with important edges. A formulation of the scores are provided in Appendix B. Note that compared to explanation evaluation with ground truths, the fidelity score is considered more faithful to the model, especially when the model makes incorrect predictions, in which case the explanation ground truths become inconsistent with evidence to making the wrong predictions. In practice, one needs to trade off between the fidelity score and the sparsity score by selecting the proper threshold for the importance.

## 4.3 QUANTITATIVE STUDIES

We conduct two groups of quantitatively experimental comparisons. We first demonstrate the explanation quality of individual tasks in terms of the fidelity score and the sparsity score. We do this by comparing TAGE with multiple baseline methods including non-learning-based methods Grad-CAM (Pope et al., 2019) and DeepLIFT (Shrikumar et al., 2017), as well as learning-based methods GNNExplainer (Ying et al., 2019) and PGExplainer (Luo et al., 2020). We do not include other optimization or search-based methods such as Monte-Carlo tree search (Jin et al., 2020) due to the

Table 2: Fidelity scores with controlled sparsity on the node-level classification dataset PPI. Each column corresponds to an explainer model trained on (or without) a specific downstream task. Underlines highlight the best explanation quality in terms of fidelity, on the same level of sparsity.

| Eval on | PGExplainer (trained on) | | | | | TAGE w/o downstream |
|---|---|---|---|---|---|---|
| | Task 0 | Task 1 | Task 2 | Task 3 | Task 4 | |
| Task 0 | **0.184 ±0.3443** | -0.005 ±0.268 | 0.033 ±0.335 | 0.034 ±0.310 | 0.018 ±0.194 | **0.271 ±0.385** |
| Task 1 | 0.046 ±0.447 | **0.197 ±0.380** | 0.043 ±0.314 | 0.008 ±0.297 | 0.021 ±0.183 | **0.300 ±0.415** |
| Task 2 | 0.028 ±0.434 | 0.001 ±0.283 | **0.345 ±0.458** | 0.024 ±0.320 | 0.097 ±0.320 | **0.499 ±0.480** |
| Task 3 | 0.075 ±0.364 | -0.015 ±0.219 | 0.036 ±0.317 | **0.262 ±0.418** | 0.040 ±0.221 | **0.289 ±0.427** |
| Task 4 | 0.035 ±0.413 | -0.021 ±0.238 | 0.223 ±0.438 | 0.075 ±0.374 | **0.242 ±0.373** | **0.330 ±0.442** |

significant time cost on real-world datasets. Note that to show the effectiveness of universal explanations over different downstream tasks, we only train one embedding explainer for all tasks in a dataset, on top of which a gradient-based downstream explainer is applied to explain multiple downstream tasks. In contrast, for existing learning-based methods, we need to train multiple explainers to explain downstream tasks individually. For all methods, we vary the threshold for selecting important nodes or edges and compare how fidelity scores change over sparsity scores on each task and dataset. The results are shown in Figure 4. In particular, TAGE outperforms other learning-based explainers on BACE, SIDER, and PPI (tasks 0 and 1). For HIV and PPI (task 2), TAGE is more effective at higher sparsity levels, *i.e.*, when fewer nodes are considered important and masked.

To justify the necessity of task-agnostic explanation and demonstrate the universal explanation ability of TAGE, we include PGExplainer as our baseline and compare the explanation quality when adopting a single explainer to explain multiple downstream tasks. For PGExplainer, we train multiple explainers on different downstream tasks and evaluate each explainer on different downstream tasks. For TAGE, we train one explainer without downstream tasks and evaluate it on different downstream tasks. Results shown in Table 2 (PPI) and Appendix D (MoleculeNet and EPN) indicate that task-specific explainers fail to generalize to different downstream tasks and hence are unable to provide universal explanations. On the other hand, the task-agnostic explainer, although trained without downstream tasks, can provide explanations with even higher quality for any downstream tasks.

GNNExplainer and PGExplainer should generally outperform task-agnostic explainers, as they are specific to data examples or tasks. This should especially be true when TAGE and PGExplainer have the same level of parameters. However, we surprisingly find that TAGE outperforms the learning-based baselines. One possible reason can be the non-injective characteristic of the downstream MLPs on top of GNNs, where different embeddings can produce similar downstream prediction results. Due to this characteristic, the learning objective of TAGE computed between embeddings brings stronger supervision than the objective computed between final predictions, as the latter objective does not guarantee consistency between embeddings or between input graphs and subgraphs.

## 4.4 MULTITASK EXPLANATION EFFICIENCY

A major advantage of the task-agnostic explanation is that it removes the need for training individual explainers, which consumes the majority of the total time cost to explain a model on a dataset. We hence evaluate the efficiency of TAGE in terms of time cost for explanation and compare it to the two learning-based explainer baselines. We record the time cost for the training and inference of different explanation methods on the same dataset and device, shown in Table 3. All results are obtained from running the explanation on the PPI dataset with 121 node classification tasks with a single Nvidia Tesla V100 GPU. Although the inference time cost of TAGE is slightly higher than that of PGExplainer, the results show TAGE costs significantly less time than GNNExplainer and PGExplainer, especially in the multitask cases ($T > 1$). TAGE allows the explanation of many downstream tasks within a reasonable time duration.

## 4.5 VISUALIZATIONS AND EXPLANATION TO EMBEDDING DIMENSIONS

We visualize the explanations of the three learning-based explanation methods on the BACE task, which aims to predict the inhibitor effect of molecules to human $\beta$-secretase 1 (BACE-1) (Wu et al., 2018). Additional visualizations on HIV and SIDER are also provided in Appendix E. The visualization results are shown in Figure 4. Each molecule visualization shows the top $10\%$ important edges

Table 3: Comparison of computational time cost among three learning-based GNN explainers on the PPI dataset. The left two columns record time cost breakdown for $T$ downstream tasks. The fourth column estimates the total time cost for explaining all 121 tasks of PPI. The last row shows the speedup times compared to GNNExplainer and PGExplainer, respectively.

| Time cost | Training (s) | Inference (s) | Total time (T=1) (s) | Est. total for 121 tasks |
|---|---|---|---|---|
| GNNExplainer | 20040.1*$T$ | – | 20040.1 | 28 d |
| PGExplainer | 7117.0*$T$ | **427.2*$T$** | 7604.2 | 10.7 d |
| TAGE | **1405.3** | 582.7*$T$ | **1988.0** | **0.83 d** |
| Speedup | **14.3*$T\times$ / 5.1*$T\times$** | $-$ / $0.73\times$ | $10.1\times$ / $3.8\times$ | $33.7\times$ / $12.9\times$ |

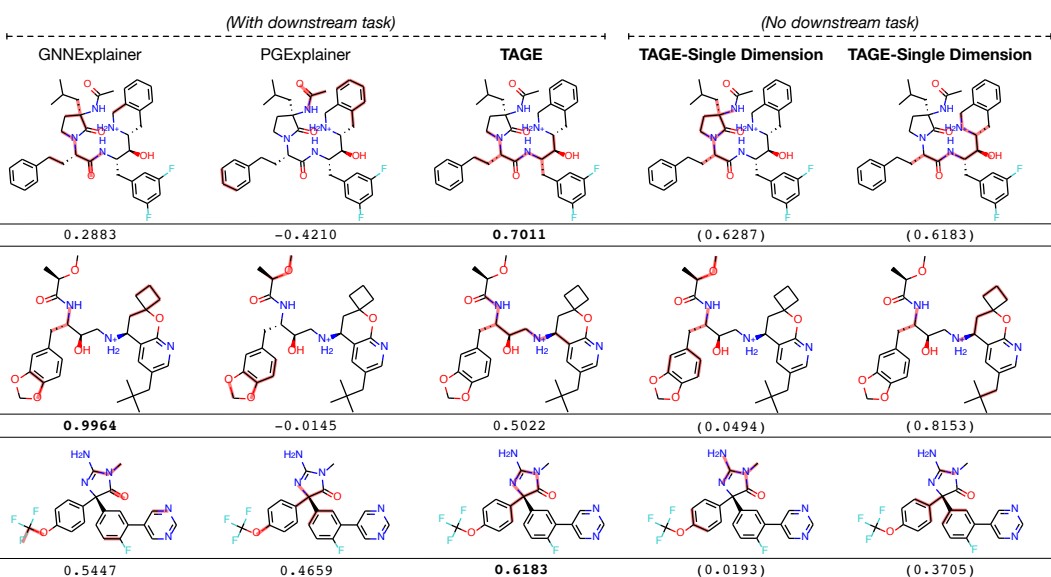

Figure 4: Visualizations on explanations to the GNN model for the BACE task. Top $10\%$ important edges are highlighted with red shadow. The numbers below molecules are fidelity scores when masking-out the top $10\%$ important edges. Right two columns are explanations to two certain embedding dimensions without downstream tasks. Fidelity scores in the right two columns explaining two embedding dimensions are still computed for the BACE task but are just for reference.

(bonds) predicted by an explainer with red shadow, together with the fidelity score on the molecule. The left three columns are explanation results with the BACE downstream task. The right columns are explanations by TAGE to two specific graph embedding dimensions, without downstream models. Embedding dimensions with greater values among all are selected in the visualizations. To obtain explanations to certain embedding dimensions, we input the one-hot vectors to the embedding explainer as condition vectors. The visualization results indicate that while baseline methods select scattered edges as important, TAGE tends to select edges that form a connected substructure, which is more reasonable when explaining molecule property predictions where a certain functional group is important for the property. In addition, the right three columns indicate that dimensions in the embedding correspond to different substructures and TAGE is able to provide explanations to the dimensions without downstream tasks.

## 5 CONCLUSIONS

Existing task-specific learning-based explainers become inapplicable under real scenarios when downstream tasks or models are unavailable and suffer from inefficiency when explaining real-world graph datasets with multiple downstream tasks. We introduced TAGE, including the task-agnostic GNN explanation pipeline and the self-supervised training framework to train the embedding explainer without knowing downstream tasks or models. Our experiments demonstrate that the TAGE generally achieves higher explanation quality in terms of fidelity and sparsity with the significantly reduced explanation time cost. We discuss potential limitations and their solutions in Appendix G.

## REPRODUCIBILITY STATEMENT

The experiment setting and evaluation protocol are described in Section 4. In particular, we describe how the models to be explained are trained. To further ensure reproducibility, we provide implementation details including explainer configurations, training settings, and how evaluation are performed in Appendix B. The detailed computation of the sparsity and fidelity scores are provided in Appendix C. The code to fully reproduce the results will be released upon acceptance.

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

## A  DATASET STATISTICS

The statistics of datasets used for evaluating TAGE are shown in Table 4.

Table 4: Statistics of multitask datasets used for explanation quality evaluation. The column "Total" under MoleculeNet indicates total number of commonly studied tasks from MoleculeNet.

|  | MoleculeNet | | | | | PPI | EPN |
|---|---|---|---|---|---|---|---|
|  | HIV | BBBP | BACE | Sider | Total | | |
| # of Graphs | 41127 | 2039 | 1513 | 1427 | – | 24 | 1 |
| Avg. # of Nodes | 25.53 | 24.05 | 34.12 | 33.64 | – | 56,944 | 5.86 mn. |
| Avg. # of Edges | 27.48 | 25.94 | 36.89 | 35.36 | – | 818,716 | 63.07 mn. |
| # of Tasks | 1 | 1 | 1 | 27 | 227 | 121 | 3 |

## B  IMPLEMENTATION DETAILS

**Structure of explainer**. Our implementation is based on Pytorch (Paszke et al., 2019), Pytorch Geometric (Fey & Lenssen, 2019), and Dive-into-graphs (Liu et al., 2021). We implement the explainer with a linear projection $f_p$ that maps the condition vector $\boldsymbol{p}$ to the same dimension as concatenated embeddings, and a 2-layer MLP with ReLU activation that maps concatenated embeddings with mask to the important score.

**Implementation of training objectives**. We adopt the Jason-Shannon Estimator as the lower bound for mutual information maximization for the two public datasets. For graph-level tasks, given a mini-batch of N samples, we consider the embeddings of a graph G and its subgraph $G_s$ as a positive pair (with N positive pairs in total), and the embeddings of a graph $G_i$ and the subgraph $G_{j,s}$ of another sample as a negative pair (with $N^2 - N$ pairs in total). For node-level tasks, we still randomly sample N nodes from the entire graph at each iteration and compute the contrastive losses on original embeddings of the $N$ nodes, and embedding of the $N$ nodes when the important subgraph is selected, respectively for each node. Similar to graph-level tasks, we consider embeddings of the same node (in original graph or in subgraph) as a positive pair, and a embeddings of node $i$ in full graph and node $j$ in selected subgraph as a negative pair ($N^2 - N$ in total).

**Training configurations**. We set the hyperparameters in the size regularization term to $\lambda_s = 0.05$ and $\lambda_e = 0.002$, respectively. For graph-level explanation on MoleculeNet, we train the embedding explainer on ZINC-2M with learning rate $1e-4$ and mini-batch size 256 for 1 epoch. The random condition vectors are generated from laplace distribution $Laplace(0, 0.2)$. For node-level explanation on PPI, we train the embedding explainer on PPI without labels with learning rate $5e-6$ and mini-batch size 4 for 1 epoch. The random condition vectors are generated from laplace distribution $Laplace(0, 0.1)$. For the EPN dataset, we train the embedding explainer with InfoNCE loss, learning rate $1e-4$ and mini-batch size 16. The random condition vectors are generated from laplace distribution $Laplace(0, 0.25)$. The hyperparameters in the size regularization term are set to $\lambda_s = 0.5$ and $\lambda_e = 0$ for the stable training with InfoNCE.

**Evaluation**. In molecule and protein property prediction, we are usually interested in the positive samples, *i.e.*, the existence of what substructure lead to a certain property. For learning-based baseline methods, we find it common that only one class of the two have good explanation, and the class with higher explanation quality is not necessarily the positive class. For example, PGExplainer has a near-to-zero fidelity score for the positive class of SIDER. We hence compare only the higher fidelity score among two classes for all explanation methods and datasets.

## C  FIDELITY AND SPARSITY

Given a set of graphs $\{G_i\}$ and node masks $\boldsymbol{m}$ predicted by the explainer, the fidelity score and the sparsity score are computed as follows.

$$Fidelity^{\,prob} = \frac{1}{N} \sum_{i=1}^{N} \left[ f(G_i)_{c_i} - f(G_i^{\mathbf{1}-\boldsymbol{m}_i})_{c_i} \right], \tag{7}$$

$$Sparsity = \frac{1}{N} \sum_{i=1}^{N} |\boldsymbol{m}_i|/|V_i|, \tag{8}$$

where $N$ denotes the number of graphs or nodes to be explained, $f$ denotes the GNN model associated with a specific downstream task, $c_i$ denotes the class of interest, which can be either the labeled class or the original predicted class, $G_i$ and $G_i^{\mathbf{1}-\boldsymbol{m}_i}$ denote the original graph and graph with important nodes removed, respectively. Explanations with both scores higher are better.

## D  ADDITIONAL RESULTS FOR UNIVERSAL EXPLANATION ABILITY

**Universal explanation performance on MoleculeNet and EPN**. Evaluation results for the universal explanation ability on the MoleculeNet and EPN datasets are shown in Table 5 and Table 6, respectively.

Table 5: Fidelity scores with controlled sparsity on graph-level molecule property prediction tasks. Each column corresponds to an explainer model trained on (or without) a specific downstream task. Underlines highlight the best explanation quality in terms of fidelity, on the same level of sparsity.

| Eval on | PGExplainer (trained on) | | | | TAGE |
|---|---|---|---|---|---|
| | BACE | HIV | BBBP | SIDER | w/o downstream |
| BACE | **0.252 ±0.340** | 0.007 ±0.251 | 0.026 ±0.022 | -0.151 ±0.330 | **0.378 ±0.293** |
| HIV | -0.001 ±0.197 | **0.473 ±0.404** | 0.013 ±0.029 | -0.060 ±0.356 | **0.595 ±0.321** |
| BBBP | 0.001 ±0.237 | -0.056 ±0.226 | **0.182 ±0.169** | -0.252 ±0.440 | **0.193 ±0.161** |
| SIDER | 0.012 ±0.219 | -0.009 ±0.212 | 0.003 ±0.029 | **0.444 ±0.391** | **0.521 ±0.278** |

Table 6: Fidelity scores with controlled sparsity on the E-commerce product dataset. Each column corresponds to one explainer model trained on different tasks or without downstream task. Underlines highlight the best explanation quality in terms of fidelity, on the same level of sparsity.

| Eval on | PGExplainer (trained on) | | | TAGE |
|---|---|---|---|---|
| | Buyers | Sellers | Reviews | w/o downstream |
| Buyers | **0.2009 ±0.2233** | 0.1731 ±0.3774 | 0.1740 ±0.4463 | **0.2713 ±0.1834** |
| Sellers | 0.5465 ±0.4773 | **0.3246 ±0.4026** | 0.1128 ±0.3019 | **0.6515 ±0.3426** |
| Reviews | 0.4178 ±0.3683 | 0.1258 ±0.3492 | **0.2310 ±0.4178** | **0.5692 ±0.4214** |

**Comparison of explanation performance when trained on different datasets**. Specifically for the MoleculeNet dataset, as there is a larger unlabeled dataset, ZINC, available for the first stage training of encoder, the training of our exlainer is also performed on the ZINC dataset. For a more strict comparison with the baseline explainer who is trained on individual MoleculeNet dataset, we additional evaluate the explanation quality when the same individual MoleculeNet dataset is used to train TAGE. The results are shown in Table 7. When trained on the same datasets individually, TAGE still performs better than the baseline explainer in terms of fidelity scores. In the individual dataset case, we need to train different explainers, similarly to the training of PGExplainer, as the datasets for the four tasks are different.

**Evaluation on the synthetic dataset BA-Shapes**. On the BACE dataset and task, we additionally compare TAGE with another recent SOTA learning-based method GEM (Lin et al., 2021) whose explainer is trained based on the Granger causality in Table 8. Note that GEM is not originally proposed under our setting. It assumes that there is a fixed number of important nodes when performing explanation and hence the final explanation is a boolean selection of nodes. We adapt GEM to compute fidelity scores under different sparsity scores by varying the threshold when generating explanation groundtruth with Granger causality.

Table 7: An ablation on training TAGE on different datasets (ZINC v.s. individual MoleculeNet datasets).

| Method | BACE | HIV | BBBP | SIDER |
|---|---|---|---|---|
| PGExplainer | 0.252 ±0.340 | 0.473 ±0.404 | 0.182 ±0.169 | 0.444 ±0.391 |
| TAGE (individual) | **0.402 ±0.281** | 0.541 ±0.330 | **0.202 ±0.157** | 0.516 ±0.292 |
| TAGE (ZINC) | 0.378 ±0.293 | **0.595 ±0.321** | 0.193 ±0.161 | **0.521 ±0.278** |

Table 8: A comparison between TAGE, GEM, and PGExplainer on BACE in terms of fidelity scores when fixing the sparsity scores. For GEM, we vary the threshold when generating explanation groundtruth with Granger causality to obtain explanations with different sparsity scores.

| Sparsity | 0.90 | 0.85 | 0.80 | 0.75 |
|---|---|---|---|---|
| TAGE | **0.3349** | **0.4992** | **0.5383** | **0.5309** |
| GEM (Lin et al., 2021) | 0.2829 | 0.3607 | 0.4260 | 0.4035 |
| PGExplainer | 0.2521 | 0.3207 | 0.4605 | 0.5161 |

# E VISUALIZATIONS ON HIV AND SIDER

Visualizations on HIV and SIDER are shown in Figure 5 and Figure 6, respectively.

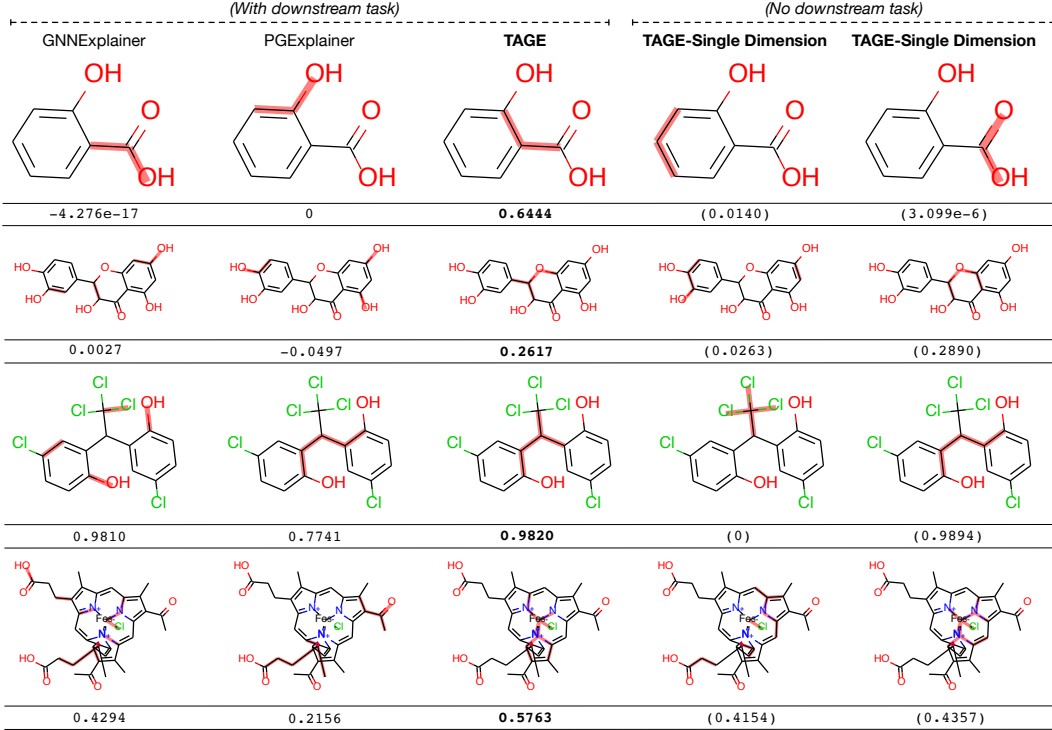

Figure 5: Visualizations on explanations to the GNN model for the HIV task. Top 10% important edges are highlighted with red shadow. The numbers below molecules are fidelity scores when masking-out the top 10% important edges. Right two columns are explanations to two certain embedding dimensions without downstream tasks.

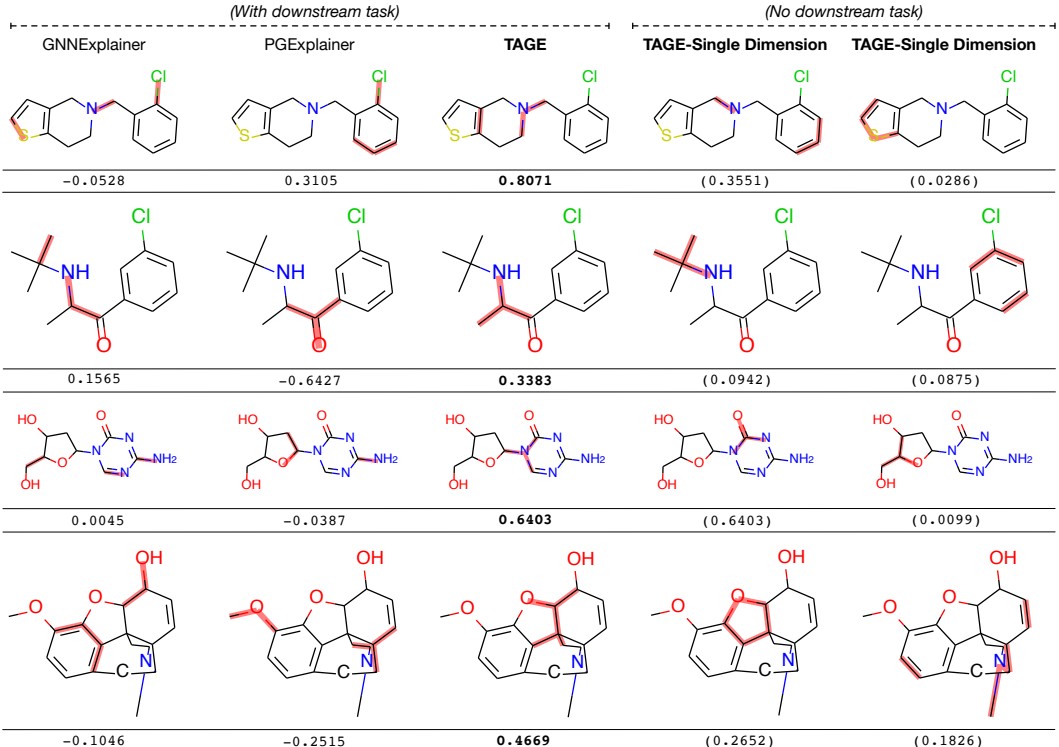

Figure 6: Visualizations on explanations to the GNN model for the SIDER task. Top 10% important edges are highlighted with red shadow. The numbers below molecules are fidelity scores when masking-out the top 10% important edges. Right two columns are explanations to two certain embedding dimensions without downstream tasks.

## F    EXPERIMENTAL STUDIES ON THE SYNTHETIC DATASETS BA-SHAPES

We perform an additional evaluation on the BA-Shapes synthetic datasets used in GNNExplainer Ying et al. (2019) and provided by Pytorch-Geometric (Fey & Lenssen, 2019). The synthetic dataset is less complicated compared to the real-world datasets. We train a 3-layer GCN for the node-classification with a training accuracy of 0.95. The AUC score (for importance edges) of TAGE is 0.999 compared to 0.963 and 0.925 of PGExplainer and GNNExplainer, respectively. Note that the baseline scores are from the PGExplainer paper and some re-implementations[2][3] of PGExplainer can also achieve an AUC score of 0.999. Our purpose to show our score on BA-Shapes is to demonstrate that TAGE is on par with its baselines even when considering the typical single-task setting. Figure 7 visualizes 20 examples of explanations. TAGE is able to provide accurate explanations for all the 20 examples.

## G    DISCUSSION OF LIMITATIONS AND POTENTIAL SOLUTIONS

**Inductive learning of explanations.** Our study focus on the setting of inductive learning of the explanation, *i.e.*, to train the explainer on a given dataset and perform inference on new coming data. There are many work conducted under the inductive setting, such as PGExplainer. All methods under this setting may have a potential limitation that the explainer may suffer from some dataset bias when training data and the data to be explained are inconsistent. This is an interesting problem that requires further investigation. However, we believe that this is a separate problem and applies

---

[2]https://github.com/LarsHoldijk/RE-ParameterizedExplainerForGraphNeuralNetworks
[3]https://openreview.net/forum?id=tt04glo-VrT

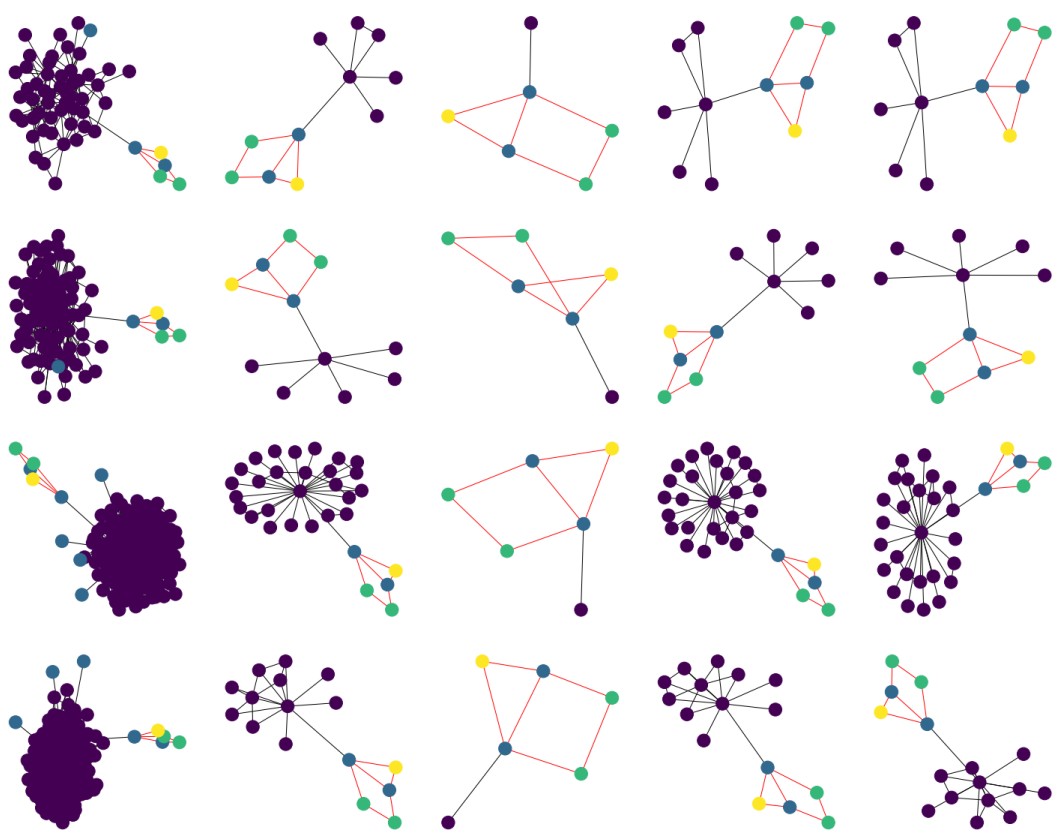

Figure 7: Visualizations on explanations to the synthetic dataset BA-Shapes.

to all inductive learning methods. In addition, the size of graph could be inconsistent for training and inference. To tackle this issue, we obtain the substructure by selecting top k percentage of edges according to their important scores.

**Black-box explanations.** Similarly to our baseline method PGExplainer, our explainer relies on node embeddings as inputs to the explainer. In particular, the node embeddings serve as representations to allow explainers identify each node. It is required by any (inductively) learning based explanations to tell neural network-based explainers which edge they are looking at. A limitation of the inductive methods is that when the node embeddings may become unavailable when explaining a black-box model. The study of explaining black-box models (where only output is available) is a different direction of study in scenarios like attacking. Many current SOTA explanation approaches, such as Grad-Cam, GNN-LRP, and PGExplainer, fail under the black-box setting. However, if one would like to adapt our approach to the black-box setting, it is still feasible by adopting a surrogate model for the black-box model and perform explanation on the surrogate model. In addition, as mentioned above, the node embeddings are mainly used to identify which node the explainer is looking at, we does not necessarily require the original embedding. When node embedding are unavailable, we can still use any representation of nodes as long as it can identify the node based on its feature and topology.

