# OpenReview forum: "Task-Agnostic Graph Neural Explanations"
_ICLR.cc/2022/Conference — ICLR 2022 Submitted_

### Official Review · Reviewer_ovjT · 2021-10-31

**Correctness:** 3
**Technical Novelty And Significance:** 2
**Empirical Novelty And Significance:** 2
**Recommendation:** 6
**Confidence:** 3

**Main Review:**

The proposed scenario where we typically need to use two-stage GNNs is interesting and makes sense.

The proposed method is simple and somehow can achieve the desired goal.

There are several concerns.

1, The setup of the explainer in this manuscript is a little different from the traditional ones. In the field of DNN (CNN or GNN) explanation, we may want to make the explanation only rely on the input sample and the model, which is most of the previous works have done. However, the proposed explainer needs one to train on the whole dataset used for the model to be explained. This may lead to several problems. 1) the explainer itself will rely on the training data, which may bring the same dataset bias to the explanation. 2) the proposed explainer, under its setup, is more like an input distillation model, where the input graph will be shrunk during the inference time.

2, The proposed method seems to be a general method for the DNN models, not limited to the GNN. I'm wondering if the authors have tried to apply the proposed method to the CNN or transformer?

3, For the evaluation, it would be interesting to apply the proposed method on the generated synthetic datasets like that used in GNNExplainer, where the ground truth explanations are available.

**Summary Of The Paper:**

In this manuscript, a new explainer for GNNs is proposed. The newly proposed method aims to provide a task-agnostic explanation for the embedding GNNs rather than a specific downstream task. The motivation of the proposed method is that the modern GNNs are typically trained in a two-stage manner, where the embedding GNN is trained in the first stage and then the task-specific lightweight MLP. The proposed method is trained based on the MI between the whole graph and sub-graph generated by the explainer.

**Summary Of The Review:**

The main concerns are about the setup of the explainer and the evaluation.

---

> ### Author Response · Authors · 2021-11-16
> **Response to comments by Reviewer ovjT**
>
> We thank the reviewer for the valuable feedback and suggestions. Below are our item-wise response to the comments.
>
> > 1, The setup of the explainer in this manuscript is a little different from the traditional ones. In the field of DNN (CNN or GNN) explanation, we may want to make the explanation only rely on the input sample and the model, which is most of the previous works have done. However, the proposed explainer needs one to train on the whole dataset used for the model to be explained. This may lead to several problems. 1) the explainer itself will rely on the training data, which may bring the same dataset bias to the explanation. 2) the proposed explainer, under its setup, is more like an input distillation model, where the input graph will be shrunk during the inference time.
>
> Our study focus on the setting of inductive learning of the explanation. There are many work conducted under the inductive setting [1, 2]. All methods under this setting may have such a limitation when training data and the data to be explained are inconsistent, but we believe this is a separate problem that requires future investigation. Regarding the concerns about the distillation, it is very common [1, 2] to obtain the most important substructure if a graph as the explanation for existing GNN explainability studies. Even if the size of graphs are inconsistent, we can obtain the substructure by selecting top k percentage of edges according to their important scores.
>
> We will include a section in Appendix G to discuss about these concerns or limitations.
>
> [1] Dongsheng Luo et al. Parameterized Explainer for Graph Neural Network. NeurIPS 2020.
>
> [2] Wanyu Lin et al. Generative Causal Explanations for Graph Neural Networks. ICML 2021.
>
>
> > 2, The proposed method seems to be a general method for the DNN models, not limited to the GNN. I'm wondering if the authors have tried to apply the proposed method to the CNN or transformer?
>
> The reviewer is right. Our studies is motivated by the two-stage setting of industrial graph data and multi-task problems in molecular biology. The proposed task-agnostic framework can be generally adapted to other domains as long as the achitectures of encoders and explainers are carefully designed for the domain. We have not performed any empirical studies on other domains, but it would be a very interesting topic to study as there are also multi-task cases for images. It is out-of-scope of the current work, but we will investigate further in this direction. Thank you for this suggestion!
>
> > 3, For the evaluation, it would be interesting to apply the proposed method on the generated synthetic datasets like that used in GNNExplainer, where the ground truth explanations are available.
>
> We thank the reviewer for the suggestion. We have performed evaluation on an additional synthetic dataset, BAShapes (and will work on more datasets). The AUC score (for importance edges) of TAGE is 0.999 compared to 0.963 and 0.925 of PGExplainer and GNNExplainer, respectively. Note that the baseline scores are from the PGExplainer paper and some re-implementations of PGExplainer can also achieve an AUC score of 0.999. The additional evaluation is only to show that TAGE is on par with its baselines even when considering the typical single-task setting. We have also included visualizations in Figure 7. All the results are in Appendix F. Code to reproduce this results are available in supplement files together with code for all our results.

---

> ### Author Response · Authors · 2021-11-23
> **Authors' follow-up**
>
> Dear reviewer ovjT,
>
> Thank you again for your valuable comments! We believe we have addressed your concerns in our previous responses. We hope that you could consider updating your score if we do have addressed your concerns. Also, please let us know if there are any additional concerns or feedback. Thank you!

---

### Official Review · Reviewer_CLGN · 2021-11-02

**Correctness:** 4
**Technical Novelty And Significance:** 3
**Empirical Novelty And Significance:** 3
**Recommendation:** 5
**Confidence:** 5

**Details Of Ethics Concerns:**

None.

**Main Review:**

**Strengths**
1.	The motivation is technically a good angle. Indeed extra costly training for generating explanations is a bottleneck for the applications.
2.	The paper is easy to understand.
3.	The method is simple but effective.

**Weakness**
1.	This paper mainly focuses on explaining multi-task models, which somehow limits the applicability.
2.	Why does the author use $\boldsymbol{p}\otimes \mathcal{E}(\mathcal{T}_\theta(\boldsymbol{p}, G))$ but $p\otimes\mathcal{E}(\mathcal{T}_\theta(\boldsymbol{p},G))$. If there is a performance gap between this two formulations, I wonder how each of them affect the quality of generated explanations.
3.	During the self-training of the embedding model, $p$ is sampled from a multivariate Laplace distribution, while later, the input is the conditional embeddings generated by the gradient. The distributions of the two groups of inputs could be different numerically and thus may affect the specific performance of the embedding model. Can the authors comment on this a bit?
4.	Some typos: last row of page 5 “as input” should be “an input”; In Section 4.1, “include four graph classiﬁcation tasks” should be “include three graph classiﬁcation tasks”.


**Summary Of The Paper:**

This paper is motivated by the fact that existing task-specific explainers are too expensive to be applied to generating explanations for a model trained for multi-tasks. They decompose the typical end-to-end learning-based GNN explainer into two parts: the embedding explainer $\mathcal{T}_{\mathcal{E}}$ and the downstream explainer $\mathcal{T}_d$ ($d$ for down).
The downstream explainer maps embeddings into importance scores $m$ for all embedding dimensions given a trained model and certain inputs. The embedding explainer is associated with the embedding model $\mathcal{E}$, which maps a given graph into a subgraph of high importance, conditioned on the embedding dimension importance.

Finally, $\mathcal{T}_{\mathcal{E}}$ is trained via a self-supervised manner, and they use a gradient explainer for $\mathcal{T}_d$. In the experiments, TAGE outperforms the SOTA explainers w.r.t fidelity, sparsity, and especially time cost.


**Summary Of The Review:**

I think this work is overall good, however, some technical details need to be clarified and the proposed model can only be necessarily applied to a limited range of explainers.

---

> ### Author Response · Authors · 2021-11-16
> **Response to comments by Reviewer CLGN**
>
> We thank the reviewer for the valuable feedback and suggestions. Below are our item-wise response to the comments.
>
> > This paper mainly focuses on explaining multi-task models, which somehow limits the applicability.
>
> The two-stage and multi-task setting are common in both industrial [1,2,3] and academic scenarios. In industrial scenarios, universal embeddings are trained in a scalable distributed setting, and used by multiple teams for their respective downstream tasks. Explanations under this setting have not been studied in the literature, since current proposed methods all require the knowledge of the downstream task and loss. Hence it is **important to develop methods specifically** for those setting.
>
> Although our study targets the difficulties in two-stage and multi-task explanations, it is **NOT limited to these settings**. When there is only one downstream task, we are still able to adopt TAGE. And our empirical studies indicates that even under the single-task settings, TAGE still have higher explanation quality compared to two learning-based baselines.
>
> Our explanation of the superior performance (even in single-task settings) of the proposed method is that optimizing mutual information computed at the embeddings can lead to stronger supervision than those computed at final predictions of the downstream task. This is due to the non-injective property of the prediction head, i.e., two embeddings with high difference may have the same predicted probability. For PGExplainer (as well as GNNExplainer), to enforce the consistency between output probabilities of the selected subgraph and original graph does not imply the consistency between embeddings, which may lead to bad explanations.
>
> [1] Priyanka Nigam et al. Semantic Product Search. KDD 2019.
>
> [2] Rex Ying et al. Graph Convolutional Neural Networks for Web-Scale Recommender Systems. KDD 2018.
>
> [3] Po-Sen Huang et al. Learning deep structured semantic models for web search using clickthrough data. CIKM 2013.
>
> > Why does the author use but. If there is a performance gap between this two formulations, I wonder how each of them affect the quality of generated explanations.
>
> Here \bm p is a randomly generated vector of dimension d to simulate the importance vector of the same dimension obtained from downstream explainer. During implementation, we sample one vector p for each iteration and apply it to all embedding in the batch. The purpose of including \bm p as a mask on embeddings is to control which dimensions the embedding explainer should explain. Replacing \bm p with a scalar p will make the conditioned training meaningless, as the explainer won’t know which dimension it should explain.
>
> > During the self-training of the embedding model, is sampled from a multivariate Laplace distribution, while later, the input is the conditional embeddings generated by the gradient. The distributions of the two groups of inputs could be different numerically and thus may affect the specific performance of the embedding model. Can the authors comment on this a bit?
>
> The use of Laplacian distribution is based on the assumption that the importance vector from downstream are sparse, i.e., only a small portion of the embedding dimensions are important. In fact, we do empirically observe a sparse distribution of values of the vectors on the datasets and tasks we study.
>
> However, when downstream model is available at the time of the training of explainer and in case we observe different or inconsistent distribution for future datasets or tasks, we can definitely adjust the distribution at training.
>
> > Some typos: last row of page 5 “as input” should be “an input”; In Section 4.1, “include four graph classiﬁcation tasks” should be “include three graph classiﬁcation tasks”.
>
> We thank the reviewer for pointing out the typos. We have updated them in our draft to avoid confusions.

---

> ### Author Response · Authors · 2021-11-23
> **Authors' follow-up**
>
> Dear reviewer CLGN,
>
> Thank you again for your valuable comments! We believe we have addressed your concerns in our previous responses. We hope that you could consider updating your score if we do have addressed your concerns. Also, please let us know if there are any additional concerns or feedback. Thank you!

---

### Official Review · Reviewer_y6T6 · 2021-11-02

**Correctness:** 3
**Technical Novelty And Significance:** 3
**Empirical Novelty And Significance:** 2
**Recommendation:** 5
**Confidence:** 5

**Main Review:**

Strengths:

1. This paper considered an interesting setting where there is no downstream task, or the target model to be explained is a multi-task model. Specifically, the entire pipeline consists of two explainers: an embedding explainer and a downstream explainer, which can be trained in isolation. The proposed method uses the downstream explainer to get dimension importance and highlight the important dimensions.

2. The presentation of the paper is in general clear.

3: It is a novel setting that is different from existing ones.


Weaknesses:

1. The technical contribution of this paper is limited. The proposed method is somewhat incremental compared with the existing method PGExplainer. Many design details are directly borrowed from PGExplainer. For example, the design of the embedding explainer and the form of the regularization term are the same.

2. There is very little intuition provided to justify the practical significance of the explanation provided by the embedding explainer if there is no downstream task. I would suggest the authors include some examples of the generated explanations, which is essential to determine whether the claimed fidelity brings meaningful results in practice.

3. Some key baselines are missing. For example, Gem [1] is also a learning-based explainer published in ICML2021. However, the author did not discuss and compare theirs with this work. I'd like to see the comparisons between the proposed method and [1].

[1] Wanyu Lin, Hao Lan, Baochun Li. Generative Causal Explanations for Graph Neural Networks. ICML 2021.


4. The proofs of Equations 2 and 3 are insufficient. Besides, in Equation 3, the definition of the positive and negative samples is unclear. The sample, positive sample, and negative sample in conditional contrastive loss should be specified for graph-level or node-level tasks.


5. Regarding the method itself, the target model of the embedding explainer seems limited to node-level embedding models instead of graph/node-level embedding models. Specifically, graph embeddings are required when training the embedding explainer. When it comes to the implementation details of embedding explainers, node embeddings are required. Although node embeddings can be simply compressed to graph embeddings, the reversion is difficult. Therefore, an embedding explainer targeting a graph-level embedding model will fail to predict node-pair importance scores.

6. "For all methods, we vary the threshold for selecting important nodes or edges and compare how fidelity scores change over sparsity scores on each task and dataset.''

How to set the threshold is not clear.

7. The empirical evidence could be more substantial.

-	In essence, an idea explainer should be human-understandable. However, by looking at the Visualization in Figure 4, model explainability is not clear. Visualizing a few instances is not convincing in terms of model explainability. Further analyses are expected.


-   About the evaluation of universal explanation ability, experiments results with downstream tasks are missing, which are indispensable for demonstrating the effectiveness of TAGE.

-   On the experimental design, some scenarios do not satisfy the multi-task setting defined in general. For example, predicting the 121 binary labels for nodes in the PPI dataset is a multi-label classification task and cannot be regarded as 121 independent tasks. It is unfair to compare with the baseline PGExplainers trained and eval on this problematic experimental setting. Therefore, the results in Table 2 do not justify the use of TAGE.

-   In Figure 3, on Task0 of the PPI data set, the fidelity scores of the proposed method are much lower than those of DeepLIFT. I would like to see a reasonable elaboration on this underperformance.


Minor points:

In figure 2, the expression of $G_s$ is wrong. It should be written as $\tau_\theta(p, G)$, the \eps operation should be done after using the GNN model.

In Section 3, $z_i$ (or $z_j$) denotes a graph embedding and a node embedding successively, which is ambiguous. If the author could distinguish between the notations of two different embeddings, this section would be more readable.


**Summary Of The Paper:**

The authors propose TAGE, a task-agnostic explanation method for explaining GNNs. TAGE explains GNN embedding models without downstream tasks and allows the explanation of multi-task models. This paper maximizes the mutual information of masked graph embedding and masked subgraph embedding as the objective function. The regularization term is added to restrict the number of edges. Empirical results focus on three aspects: (1) Improvement in fidelity scores brought by TAGE; (2) Visualizations on explanations to the GNN model; and (3) Comparison of computational time cost among GNN explainers.


**Summary Of The Review:**

The technical contribution of this paper is limited. The proposed method is somewhat incremental compared with the existing method PGExplainer.  The paper lacks rigorous empirical analysis, especially in terms of studying the interpretability aspect.

---

> ### Author Response · Authors · 2021-11-16
> **Response to comments by Reviewer y6T6 (con'd 2)**
>
> > About the evaluation of universal explanation ability, experiments results with downstream tasks are missing, which are indispensable for demonstrating the effectiveness of TAGE.
>
> All of our quantitative evaluations are performed on individual downstream tasks as the fidelity score cannot be computed without downstream predictions. In Tables 2, 5, and 6, each column correspond to a explainer trained with one specific task or trained without downstream tasks, and each row shows the results of evaluating the explainers on individual downstream tasks. We have updated certain parts of our draft to make them more clear.
>
> > On the experimental design, some scenarios do not satisfy the multi-task setting defined in general. For example, predicting the 121 binary labels for nodes in the PPI dataset is a multi-label classification task and cannot be regarded as 121 independent tasks. It is unfair to compare with the baseline PGExplainers trained and eval on this problematic experimental setting. Therefore, the results in Table 2 do not justify the use of TAGE.
>
> Our study focuses on the new two-stage and multi-task settings. Currently, there are no learning-based explanation method directly applicable to such settings. We adapt GNNExplainer and PGExplainer to the settings we study as our baselines.
>
> We argue that the 121 binary labels in PPI is exactly a multitask case where previous methods failed to efficiently provide explanations. Each label corresponds to one specific protein property as a prediction task. Although one can train only one prediction model (with 121 output dimensions) for PPI, PGExplainer can **only explain one output dimension at a time** due to its task-specific design. In other words, the PGExplainer is not able to take the 121 dimensional predictions as its input, and output 121 subgraphs (one subgraph explanation to each prediction target). As a result, 121 individual explainers are required to be trained in order to explain the 121 prediction targets.
>
> > In Figure 3, on Task0 of the PPI data set, the fidelity scores of the proposed method are much lower than those of DeepLIFT. I would like to see a reasonable elaboration on this underperformance.
>
> The results on task 0 of PPI are very interesting and surprising as it is the only task that DeepLIFT outperforms all the learning-based explanation methods. We believe that it is possible that DeepLIFT can be specifically suitable on explaining certain tasks and datasets and PPI-0 may be one of them. However, when comparing among learning-based methods, TAGE is still at the same level or better than GNNExplainer and PGExplainer, which are closer baselines of TAGE. And the performance of TAGE is quite consistent among different tasks and datasets. Therefore, we would rather consider it as a surprising outlier of DeepLIFT than an underperformance of TAGE.

---

> ### Author Response · Authors · 2021-11-16
> **Response to comments by Reviewer y6T6 (con'd)**
>
>
>
> > The proofs of Equations 2 and 3 are insufficient. Besides, in Equation 3, the definition of the positive and negative samples is unclear. The sample, positive sample, and negative sample in conditional contrastive loss should be specified for graph-level or node-level tasks.
>
> We thank the reviewer for pointing out the unclear parts. JSE and InfoNCE are two common upper bounds to the mutual information. Their comprehensive proofs of the two equations can be found in [4] (JSE) and [5] (InfoNCE), respectively.
>
> For graph-level tasks, given a mini-batch of N samples, we consider the embeddings of a graph G and its subgraph G_s as a positive pair (with N positive pairs in total), and the embeddings of a graph G_i and the subgraph G_{j,s} of another sample as a negative pair (with N^2-N pairs in total). For node-level tasks, we still randomly sample N nodes from the entire graph  at each iteration and compute the contrastive losses on original embeddings of the N nodes, and embedding of the N nodes when the important subgraph is selected, respectively for each node. Similar to graph-level tasks, we consider embeddings of the same node (in original graph or in subgraph) as a positive pair, and a embeddings of node i in full graph and node j in selected subgraph as a negative pair (N^2-N in total).
>
> We have updated our draft (Appendix B) to make the above details more clear.
>
> [4] R Devon Hjelm et al. Learning deep representations by mutual information estimation and maximization. ICLR 2019.
> [5] A Oord et al. Representation Learning with Contrastive Predictive Coding. Preprint 2019.
>
> > Regarding the method itself... Therefore, an embedding explainer targeting a graph-level embedding model will fail to predict node-pair importance scores.
>
> In fact, the study of explaining black-box models (where only output is available) is a different direction of study in scenarios like attacking. Many current SOTA explanation approaches, such as Grad-Cam, GNN-LRP, and PGExplainer, fail under the black-box setting. Similarly, our study mainly focus on cases when the GNN model is available to study, i.e., we are able to access its gradients, node embeddings before readout, etc., which are common in scenarios like molecule chemical research and industrial graph data mining. In particular, the node embeddings serve as representations to allow explainers identify each node. It is required by **any (inductively) learning based explanations** to tell neural network-based explainers which edge they are looking at.
>
> However, if one would like to adapt our approach to the black-box setting, it is still feasible by adopting a surrogate model for the black-box model and perform explanation on the surrogate model. In addition, as mentioned above, the node embeddings are mainly used to identify which node the explainer is looking at, we does not necessarily require the original embedding. When node embedding are unavailable, we can still use **any representation of nodes as long as it can identify the node** based on its feature and topology.
>
> We have updated our draft to include the above discussions in Appendix G.
>
> > "For all methods, we vary the threshold for selecting important nodes or edges and compare how fidelity scores change over sparsity scores on each task and dataset.'' How to set the threshold is not clear.
>
> We vary the threshold by setting the percentage (2%, 5%, 10%, etc.) of selected important edges based on their scores, i.e., edges with top 2/5/10% highest score are considered in the important subgraph when computing the scores. The sparsity of selecting the top 2% important edges are higher than the sparsity of selecting the top 5%.
>
> > The empirical evidence could be more substantial... Further analyses are expected.
>
> Besides Figure 4, we have provided additional visualization in appendices (Figures 5 and 6). In terms of the evaluation of explanation quality, the fidelity score is currently the most reliable evaluation metric, as it provide faithful measurement to model decisions no matter the model itself is good or bad. Sometimes the model decisions themselves are anti-intuition. In those cases, we should focus more on the faithfulness of explanation to the model than the human intuition of an explanation. In fact, it is exactly what deep model explanations aim to provide.
>
>
> Our evaluation protocols including fidelity scores and visualizations are commonly used in previous works such as [6]. However, to make the evaluation more convincing, we have performed an additional experimental study on the synthesis BAShapes whose the ground truths are available. We included the results in Appendix F. Code to reproduce this results are available in supplement files together with code for all our results.
>
>
> [6] Hao Yuan et al. On Explainability of Graph Neural Networks via Subgraph Explorations. ICML 2021.

---

> ### Author Response · Authors · 2021-11-16
> **Response to comments by Reviewer y6T6**
>
> We thank the reviewer for the valuable feedback and suggestions. Regarding the reviewer’s concerns on the technical contribution, we would like to first highlight our novelty and contributions.
>
> Our work studies a novel explanation problem under the **two-stage and multi-task learning** settings. The settings are widely adopted in the industry [1,2,3] where universal embeddings are trained in a scalable distributed setting, and used by multiple teams for their respective downstream tasks. Explanations under this setting have not been studied in the literature, since current proposed methods all require the knowledge of the downstream task and loss. Regarding the techniques, we propose a **novel conditional objective** based on mutual information, cooperating with the conditional explainer, to enable diverse explainability conditioned on different downstream tasks.
>
> [1] Priyanka Nigam et al. Semantic Product Search. KDD 2019.
> [2] Rex Ying et al. Graph Convolutional Neural Networks for Web-Scale Recommender Systems. KDD 2018.
> [3] Po-Sen Huang et al. Learning deep structured semantic models for web search using clickthrough data. CIKM 2013.
>
> Below are our item-wise response to the comments.
>
> > The technical contribution of this paper is limited. The proposed method is somewhat incremental compared with the existing method PGExplainer. Many design details are directly borrowed from PGExplainer. For example, the design of the embedding explainer and the form of the regularization term are the same.
>
> PGExplainer and our work study different problems. Our work is orthogonal to the direction of designing novel backbone explainer architectures.
>
> As mentioned above, our work targets the **novel task-agnostic explanation problem**. Current explainers are inapplicable for the two-stage setting or extremely inefficient for the multi-task setting. On the technical front, we propose the task-agnostic explanation framework, in which we design the novel conditional objective, conditional explainer, and training framework. Our method allows the training of explanations under a more difficult, strict, but common problem setting without access to downstream tasks.
>
>
> > There is very little intuition provided to justify the practical significance of the explanation provided by the embedding explainer if there is no downstream task. I would suggest the authors include some examples of the generated explanations, which is essential to determine whether the claimed fidelity brings meaningful results in practice.
>
> Please note that **only the training** of our explainer does not require downstream tasks. All the **evaluations** of our methods are performed **on downstream tasks**. To justify the practical significance, all our evaluations are performed on real-world datasets and tasks motivated by real problems on graphs. In addition, visualizations (Figures 4, 5, and 6, column 3) show examples of explanations generated by TAGE on individual downstream task, i.e., BACE, HIV, and SIDER.
>
> We highlight that we are not claiming that the explanations are meaningful without downstream tasks. We have updated certain parts in our draft to avoid this confusion.
>
>
> > Some key baselines are missing. For example, Gem [1] is also a learning-based explainer published in ICML2021. However, the author did not discuss and compare theirs with this work. I'd like to see the comparisons between the proposed method and [1].
>
> We highlight that our study focuses on a **new problem setting** and there are no baseline methods directly applicable under this setting. Gem[1], similar to PGExplainer, provides explanations under the conventional setting where the tasks are known while training. We will include the results with Gem once they are available.

---

> ### Author Response · Authors · 2021-11-20
> **Comparison results with GEM**
>
> Dear reviewer,
>
> Thank you again for your feedback and suggestions. As suggested, we additionally compare the explanation quality with GEM on the BACE task (as a single-task setting). As GEM is not originally proposed under the setting to compute important scores of nodes/edges (it performs a binary selection of fixed number of nodes), we adapt GEM to compute fidelity scores under different sparsity scores by varying the threshold when generating explanation ground-truth with Granger causality. Below are the results.
>
> |Sparsity|0.90|0.85|0.80|0.75|
> |---|---|---|---|---|
> |TAGE|0.3349|0.4992|0.5383|0.5309|
> |GEM|0.2829|0.3607|0.4260|0.4035|
> |PGE|0.2521|0.3207|0.4605|0.5161|
>
> We have included the results and detailed descriptions in Appendix D.

---

> ### Author Response · Authors · 2021-11-23
> **Authors' follow-up**
>
> Dear reviewer y6T6,
>
> Thank you again for your valuable comments! We believe we have addressed your concerns in our previous responses. We hope that you could consider updating your score if we do have addressed your concerns. Also, please let us know if there are any additional concerns or feedback. Thank you!

---

### Official Review · Reviewer_QC3Y · 2021-11-03

**Correctness:** 4
**Technical Novelty And Significance:** 3
**Empirical Novelty And Significance:** 3
**Recommendation:** 5
**Confidence:** 3

**Main Review:**

The paper provides an interesting framework to tackle graph explanation.

(1) Competitive baseline

Other methods, such as PGExplainer, are training multiple explainers on different downstream tasks and evaluating each explainer on different downstream tasks. Should they have more expression power to achieve an upper bond for one universal explainer for all tasks or at least can recover the results of one universal explainer?  However, existing methods showed much worse results in Fig3. I speculate the reason maybe that many aspects of the two methods are different. Therefore, it will be interesting to see under exactly the same setup of Task-Agonistic, but one trainer per task, if the proposed method doesn't lead to much information drop then it demonstrates the good performance.

(2) Novelty
The paper is quite similar with graph pretraining to learn a useful embedding and contrastive learning with mutual information , but with additional explanation part. Can the author recap on the major novelty contribution?

(3) Intuition of design of p
Page 4 sec 3.2 "We introduced a masking vector p  to indicate specific dimensions of embeddings on which to maximize MI. During
explanation, we obtain the masking vector from the importance vector computed by any downstream
explainer T_down. As no downstream importance vector is available at training, we sample the masking vector p randomly".

How can the randomly selection of p during training help with specific dimension of embedding from importance vector during testing?



**Summary Of The Paper:**

The authors proposed a method to offer explanations for a multitask prediction model with a single explainer. The method provides explanations in cases where the GNN is trained in a self-supervised manner, and the resulting representations are used in future downstream tasks.


**Summary Of The Review:**

The paper is tackling an interesting problem -- explanation of graph. However, some results are a bit counter-intuitive.

---

> ### Author Response · Authors · 2021-11-16
> **Response to comments by Reviewer QC3Y**
>
> We thank the reviewer for the valuable feedback and suggestions. Below are our item-wise response to the comments.
>
> > (1) Competitive baseline: ... Therefore, it will be interesting to see under exactly the same setup of Task-Agonistic, but one trainer per task, if the proposed method doesn't lead to much information drop then it demonstrates the good performance.
>
> The proposed task-agnostic explainer naturally does not require downstream tasks during training by design. For the PPI and EPN dataset, the data are **exactly the same** for different downstream tasks. In this case, even if we want to train an individual TAGE explainer per task, we are **repeatedly training the same explainer under exactly the same setting multiple times** and the explainers do not have any difference except the random initialization of their parameters. Therefore, the comparison between TAGE and PGExplainer trained with any tasks is exactly what the reviewer suggests. The results show TAGE still outperforms PGExplainer when considering only one task.
>
> For the MoleculeNet dataset, as the encoder is pre-trained on ZINC whereas the downstream task **uses different datasets** from MoleculeNet, the dataset for training are different for PGExplainer and TAGE. In this case, task-agnostic explainers are trained on the pre-training dataset, i.e., ZINC, whereas task-specific explainers are trained on datasets for individual downstream tasks. In this case, there is a difference between the two training settings as mentioned by the reviewers. Therefore, as suggested, we conduct additional experiments and train individual TAGE on each downstream dataset. The results (fidelity scores at the same level of sparsity) are shown in the table below.
>
> |Method|BACE|HIV|BBBP|SIDER|
> |---|---|---|---|---|
> |PGExplainer|0.252 ±0.340|0.473 ±0.404|0.182 ±0.169|0.444 ±0.391|
> |TAGE (individual downstream)|0.402 ±0.281|0.541 ±0.330|0.202 ±0.157|0.516 ±0.292|
> |TAGE (ZINC)|0.378 ±0.293|0.595 ±0.321|0.193 ±0.161|0.521 ±0.278|
>
> The additional results indicate that when training TAGE on individual downstream datasets, the performance in terms of fidelity score is similar to TAGE trained on ZINC (two better and two worse) and is still better than the baseline method.
>
> Our explanation of the superior performance of our proposed method is that optimizing mutual information computed at the embeddings can lead to stronger supervision than those computed at final predictions of the downstream task. This is due to the non-injective property of the prediction head, i.e., two embeddings with high difference may have the same predicted probability. For PGExplainer (as well as GNNExplainer), enforcing consistency between output probabilities of the selected subgraph and original graph does not imply consistency between embeddings, which may lead to bad explanations.
>
> > (2) Novelty: ... Can the author recap on the major novelty contribution?
>
> Our work studies explanations for the **two-stage and multi-task** settings. The settings are widely adopted in the industry where universal embeddings are trained in a scalable distributed setting, and used by multiple teams for their respective downstream tasks [1,2,3]. Explanations under this setting have not been studied in the literature, since current proposed methods all require the knowledge of the downstream task and loss. To address the problem, we are the first to propose the task-agnostic framework. Regarding the techniques, we propose a **novel conditional objective** based on mutual information, cooperating with the conditional explainer, to enable diverse explainability conditioned on different downstream tasks.
>
> [1] Priyanka Nigam et al. Semantic Product Search. KDD 2019.
>
> [2] Rex Ying et al. Graph Convolutional Neural Networks for Web-Scale Recommender Systems. KDD 2018.
>
> [3] Po-Sen Huang et al. Learning deep structured semantic models for web search using clickthrough data. CIKM 2013.
>
> > (3) Intuition of design of p: ... How can the randomly selection of p during training help with specific dimension of embedding from importance vector during testing?
>
> We assume that the importance vectors from downstream tasks are sparse, i.e., only a small portion of the embedding dimensions are important. In fact, we do empirically observe a sparse distribution of values of the vectors and that only the sparsity pattern (but not the scale) affect the explanation results. Therefore, we sample the condition vectors from a Laplacian distribution to simulate the importance vector obtained from a random downstream task and a random data sample. Note that different vectors are generated at each iteration of training. **The randomness during training allows generated vectors to cover more possible importance vectors during inference for higher generalizability of the explainer.** For example, an importance vector [0,0,0.8,0.1] may have a similar vector occurred during training when there are enough iterations.

---

> ### Author Response · Authors · 2021-11-23
> **Authors' follow-up**
>
> Dear reviewer QC3Y,
>
> Thank you again for your valuable comments! We believe we have addressed your concerns in our previous responses. We hope that you could consider updating your score if we do have addressed your concerns. Also, please let us know if there are any additional concerns or feedback. Thank you!

---

### Author Response · Authors · 2021-11-21
**Summary of authors responses. Please let us know if any questions or additional comments.**

Dear reviewers and area chair,

We thank all the reviewers for taking time to provide insightful comments and constructive suggestions. They are very helpful for us to strengthen the submission.

The reviewers agree that our problem is interesting and novel, the proposed method is effective, and our paper is overall well-written. Regarding the concerns and questions raised by the reviewers, we have provided detailed responses and updated our draft accordingly to refine our presentation and address the concerns. Given your overall positive comments, we hope that you could update your evaluation if we have addressed your concerns.

To summarize our responses, besides answering questions and refining our representation, I) we highlight the significance and technical contributions of our work. In particular, we study a **novel and important explanation problem** under commonly adopted settings for real-world tasks, which, however, has not been studied in the literature. Technically, we proposed the **novel MI-based conditional objective, the conditional explainer, and the self-supervised training framework for the explainer** to enable diverse explainability conditioned on different downstream tasks. II) We conduct **additional experiments and ablations** including 1) the comparison with more recent explanation approach GEM, 2) the comparisons on MoleculeNet with training TAGE on different datasets for individual task (same setting of training task-specific explainers), and 3) additional results on the synthetic dataset BA-Shapes where the ground truths are available.

We would like to know if our response has properly addressed your concerns. Please let us know if you have any questions or additional feedback.

---

### Author Response · Authors · 2021-11-28
**Authors' follow-up**

Dear reviewers,

Since the discussion period will end soon, could you kindly check our response and revision? We believe we have addressed all of your concerns and are looking forward to hearing from you.

Thank you!

---

### Decision · Program_Chairs · 2022-01-20

**Decision:**

Reject

**Comment:**

Summary of the paper and the reviews:
The authors propose a method to explain the GNN predictions in a task-agnostic setting, meaning that the method can be applied to a new downstream task without fine-tuning. The task is formulated as to predict the important subgraph given the input graph and the ground truth label. The learning algorithm optimizes the mutual information between the embedding of the subgraph and the original graph. The experiment shows the quantitative improvement measured by the fidelity score, the qualitative visualization of highlighted subgraphs and comparison of the cost w.r.t the baseline GNN explainer models.
Strength:
1) The task-agnostic setting is novel.
2) The proposed method shows improvement in the fidelity score with a reduced training cost over the baseline models
Weakness:
1) The proposed objective requires additional justification. To optimize the intractable mutual information objective, the authors propose JSE and InfoNCE as upper bound estimations of mutual information, but the negative sampling technique in the proposed method is not fully justified.
2) During training, the authors simulate the task-specific importance vector by sampling masks from a Laplace distribution. During testing, the importance vector is obtained by gradient-based approach. Further analysis is needed to quantify the effect of this training/testing discrepancy.
3) In the empirical experiment, the proposed task-agnostic model outperforms the task-specific baselines. Why should such an outcome happen? The reason needs additional analysis but is not provided in the current paper. Moreover, the qualitative results of a few examples are not sufficiently convincing for the reported empirical success.

Summary of the discussions and the decision by reviewers: One reviewer asked for a justification of the negative sampling approaches used to approximate the mutual information objective. While the authors described their implementation design in their rebuttal, the theoretical justification of the method was not enough. Two reviewers raised the question about how randomly sampling importance masks during training could affect the downstream tasks performance, which was not fully addressed in the rebuttal. Other than that, the experimental concerns about new baselines and datasets were well addressed by the authors.

Recommendation: The paper has received borderline review scores (5, 5, 5, 6). Although the authors addressed some of the concerns in their rebuttal on the experimental design and added important baselines, more convincing justifications/analysis for the proposed method are still missing. Therefore, the reviewers didn’t raise their scores. Based on the above concerns, the recommendation is to reject.